# Neurons from human mesenchymal stem cells display both spontaneous and stimuli responsive activity

Nihal Karakaş[1,2]*, Sadık Bay[2], Nezaket Türkel[3], Nurşah Öztunç[2,4], Merve Öncül[2], Hülya Bilgen[5], Khalid Shah[6], Fikrettin Şahin[3], Gürkan Öztürk[2,7]

1 Medical Biology Department, School of Medicine, İstanbul Medipol University, İstanbul, Turkey,
2 Regenerative and Restorative Medicine Research Center (REMER), Research Institute for Health Sciences and Technologies (SABITA), İstanbul Medipol University, İstanbul, Turkey, 3 Genetics and Bioengineering Department, Faculty of Engineering, Yeditepe University, İstanbul, Turkey, 4 Medical Biology and Genetics Program, Graduate School of Health Sciences, İstanbul Medipol University, İstanbul, Turkey, 5 Center for Bone Marrow Transplantation, İstanbul Medipol University Hospital, İstanbul, Turkey, 6 Center for Stem Cell Therapeutics and Imaging, Brigham and Woman's Hospital, Harvard Medical School, Boston, Massachusetts, United States of America, 7 Physiology Department, International School of Medicine, İstanbul Medipol University, İstanbul, Turkey

☯ These authors contributed equally to this work.
* nkarakas@medipol.edu.tr

**Data Availability Statement:** Data cannot be shared publicly because of patient privacy inquiries in written consents of donors whose bone marrow samples used in this study. Data are available from

## Abstract

Mesenchymal stem cells have the ability to transdifferentiate into neurons and therefore one of the potential adult stem cell source for neuronal tissue regeneration applications and understanding neurodevelopmental processes. In many studies on human mesenchymal stem cell (hMSC) derived neurons, success in neuronal differentiation was limited to neuronal protein expressions which is not statisfactory in terms of neuronal activity. Established neuronal networks seen in culture have to be investigated in terms of synaptic signal transmission ability to develop a culture model for human neurons and further studying the mechanism of neuronal differentiation and neurological pathologies. Accordingly, in this study, we analysed the functionality of bone marrow hMSCs differentiated into neurons by a single step cytokine-based induction protocol. Neurons from both primary hMSCs and hMSC cell line displayed spontaneous activity ($\geq$75%) as demonstrated by $Ca^{++}$ imaging. Furthermore, when electrically stimulated, hMSC derived neurons (hMd-Neurons) matched the response of a typical neuron in the process of maturation. Our results reveal that a combination of neuronal inducers enhance differentiation capacity of bone marrow hMSCs into high yielding functional neurons with spontaneous activity and mature into electrophysiologically active state. Conceptually, we suggest these functional hMd-Neurons to be used as a tool for disease modelling of neuropathologies and neuronal differentiation studies.

## Introduction

Mesenchymal stem cells (MSCs) are heterogeneous population of multipotent and committed progenitor cells that can differentiate to end-stage lineage cells including osteoblasts, chondrocytes, adipocytes, muscle cells, pericytes, reticular fibroblasts, and even neurons [1–5]. MSCs

Ethics Committee (contact via ilknurfil@medipol.edu.tr) for researchers who meet the criteria for access to confidential data.

**Funding:** This study was supported by İstanbul Medipol University Scientific Research Projects Committee with project number 2019-14 (N.K.) and Yeditepe University, Genetics and Bioengineering Department (N.T.).

**Competing interests:** The authors have declared that no competing interests exist.

exist in bone marrow, adipose tissue, amniotic fluid, dental pulp, endometrium, muscle, periosteum, placenta, synovial fluid and Wharton's jelly [6]. MSCs have some superior properties compared to other stem cell types. These properties are easy to isolate and expand, high differentiation capacity, low immunological response, low risk of tumour formations and giving ethical permissions easily [7,8]. As a further step, the potential use of MSCs has been reported in clinical trials that are focused on treating several diseases such as myocardial infarction, stroke, Crohn's disease, and diabetes [7]. MSCs are studied for central nervous system and peripheral nervous system repairments [9]. For instance, differentiated functional neurons from MSCs were used in the studies targeting neurodegenerative diseases [10,11].

Various strategies have been developed over the years to transdifferentiate MSCs into neurons. Inducers used in these strategies are generally divided into four groups: small molecules, epigenetic modifications, psychotropic drugs and enriched media (medium coctails with chemical inducers) [12]. Alternatively, differentiation of neurons from MSCs was reported in different ways such as using co-culture conditions, exosomes and graphene substrates [13–15]. Neuronal differentiation attempts of MSCs in media containing chemical reagents provided a faster differentiation rate compared to other methods [16]. The most accaptable identifiers of neuronal differentiation are morphological changes and neuronal gene expression profiles. Morever, the functionality of differentiated neurons is one of the misleading fact that has to be examined for each newly defined protocols [17].

To evaluate the functionality of neurons from hMSCs, electrophysiological recordings and calcium imaging can be applied and then analysed for the activity pattern of neurons from hMSCs as compared to primary neurons [18]. Among neuronal activity evaluation methods, patch-clamp and fluorescent imaging of calcium ion transport are quite accurate techniques to record and visualize the synaptic currents [19,20]. Another nature of most primary neurons is their ability to generate signals without any outside stimulations. This is generally termed as spontaneous activity and in neurons indicates that the neuronal network is developing and besides it is important for cellular specificity [20]. Since human MSCs are able to transdifferentiate into neurons following several ways, of the utilization of hMSC derived neurons within a clinical aspect, functionality has become important in neuronal differentiation studies [21–23]. Only few studies have shown that human mesenchymal stem cell (hMSC)-derived neurons are functionally active upon chemical or electrical excitation [24]. Other methods might be either not evaluated or the results of neuronal activity studies were negative. Accordingly, prior to use of hMSC derived neurons as a tool for disease modeling or regenerative applications, the activity pattern and functional synapse formations of these neurons have to be defined to move the neuronal differentiation attempts a step further [25–27].

In this study, hMSC cell line and hMSCs isolated from healthy donors were differentiated into neurons using media that contains a combination of certain neuronal differentiation and survival inducers. We demonstrated that differentiated hMSCs expressed neuronal markers including synaptic proteins. Our study revealed the ability of bone marrow hMSC derived neurons (hMd-Neurons) can generate spontaneous activity and respond to electrical stimulation. Herein, we introduce hMSC-derived neurons (hMd-Neurons) with high functional neuron ratios in a defined induction media that can be further studied and potentially link the technical bridges to broadly study the nervous system deficiencies in several aspects.

## Materials and methods

### Cell lines and reagents

Human bone marrow derived mesenchymal stem cell (hMSC) line (UE7T-13 cells, no. RBRC-RCB2161; RIKEN, Japan) [28–32], Growing and expansion media of hMSCs:

Dulbecco's Modified Eagle's Medium (DMEM; Gibco) with 2mM $_L$-Glutamine, Fetal bovine serum (FBS; Gibco), Penicillin/Streptoycin (Gibco). Neuronal induction: dibutyryl cyclic AMP (dbcAMP; Sigma), 3-isobutyl-1-methylxanthine (IBMX; Sigma), human epidermal growth factor (hEGF; Sigma), recombinant human basic fibroblast growth factor (bFGF; R&D systems), fibroblast growth factor-8 (FGF-8; Pepro Tech), recombinant human brain-derived neurotrophic factor (BDNF; R&D systems), and nerve growth factor (NGF), Neurobasal medium (Gibco) supplemented with 2% B27 supplement (Gibco), 2 mM $_L$-glutamine (Gibco).

## Sample collection and ethical issues

Human bone marrow aspirates of healthy donors were supplied by Istanbul Medipol University, Center for Bone Marrow Transplantation. Bone marrow samples (n = 5, 1 female and 4 males) were collected from healthy donors from ages 2–18 and used in this study. Written consents of donors (parents of the matched allogeneic donors) were obtained, documented and witnessed. While doing the harvesting of the bone marrow for transplantation, 20 ml/ kg of donor bone marrow was harvested (400–1000 ml) and 5 ml was kept for the study. The 395–995 ml was infused to the patient. All procedures were approved by Ethics Committee of Author University (no.425 on 10.25.2017).

## Isolation and expansion of bone marrow mesenchymal stem cells

Human mesenchymal stem cells (hMSCs) were isolated from bone marrow by ficoll density gradient centrifugation. After isolation, cells were cultured in Dulbecco's Modified Eagle Medium-low glucose (DMEM-LG, Gibco) culture medium containing 20% fetal bovine serum (FBS, Gibco) and 0.2% primocin (Gibco). Cells were incubated at 37°C in a humidified 5% $CO_2$ chamber. Then, non-adherent cells were removed via medium refreshment after 3 days (d) and adherent cells were labeled as passage 0 (P0) and grown to 80% confluence. When the cells reached 80% confluence, they were detached with 0,25% Trypsin/EDTA solution. The cells were harvested and subcultured at a density of 1.5 x$10^3$ cells/cm$^2$ in culture flasks per 5–6 days. Flow cytometry analysis was performed at passage 3 (P3) and each donor derived hMSCs were processed in the same order and used for all experiments.

## Flow cytometry

To confirm hMSC phenotype of isolated cells grown in culture, cells were subjected to flow cytometry analysis. Flow cytometry was performed using a FACS (BD Influx Cell Sorter with Bioprotect IV Safety Cabinet) system. The data were analysed with FlowJo software (Treestar) and the forward and side scatter profile gated out debris and dead cells. Immunophenotyping of human BM-MSCs was performed with antibodies against the following human antigens: CD14 (Abcam; ab82434), CD29 (Biolegend; 303004), CD31 (Abcam; ab27333), CD34 (Abcam; ab18227), CD44 (Abcam; ab58754), CD45 (Abcam; ab134202), CD73 (Abcam; ab157335), CD105 (Abcam; ab53321), and their isotype controls. For determining whether proliferating stem/progenitor cells are present among hMd-Neurons, we analysed cell fractions of hMd-Neurons by Nestin (BD; 51–9007230), Ki-67 (BD; 51–9007231) and Sox-2 (BD; 51–9007227) antibodies.

## Mesodermal differentiation

To qualify isolated cells from human BM as hMSCs, cells were induced to differentiation into adipogenic, osteogenic and chondrogenic lineages. Briefly, for adipogenic differentiation, 5x$10^3$ hMSCs/cm$^2$ were exposed to Complete MesenCult Adipogenic Medium containing

MesenCult MSC Basal Medium (Stemcell) and 10% Adipogenic Stimulatory Supplement (Stemcell) for 3 weeks. For osteogenic differentiation, $2x10^5$ cells/cm$^2$ incubated with Complete MesenCult Osteogenic Medium including MesenCult MSC Basal Medium, Osteogenic Stimulatory Supplement, β-Glycerophosphate, Dexamethasone, Ascorbic acid (all from Stemcell) for 5 weeks. For chondrogenic differentiation, $7.5x10^6$ cells/cm$^2$ were cultured for 3 weeks with Stempro Chondrocyte Differentiation Basal Medium (Gibco) containing 10% Stempro Chondrogenesis Supplement (Gibco). Standard histochemical staining methods were applied. Osteogenic, adipogenic and chondrogenic differentiation were confirmed by Toluidine Blue, Oil Red O, Alcian Blue staining, respectively.

## Neuronal induction

For neuronal induction experiments, we used a commercial hMSC cell line (UE7T-13 cells, no. RBRC-RCB2161; RIKEN, Japan) and hMSCs isolated from healthy donors (n = 5). We repeated experiments for each hMSC donors used in this study. hMSCs at P3 were seeded on culture dishes prior to neuronal induction. Cell density was optimized to $3,0x10^3$. Culture was maintained via using DMEM containing 10% FBS + 1% Penicillin/Streptomycin and cells were incubated in 37˚C, 5% $CO_2$ incubator for 24 hrs. Neuronal induction media was prepared with 20 ng/ml hEGF, 40 ng/ml bFGF, 10 ng/ml FGF-8, 10 ng/ml human BDNF, 40 ng/ml NGF, 0.125 mM dbcAMP, 0.5 mM IBMX, 2 mM L-glutamine in Neurobasal medium + B27 supplement in the absence of serum. Cells were then treated with neuronal induction media by medium refreshment every 48 hours throughout the differentiation process. In line with this, samples were collected for reverse transcriptase PCR (RT-PCR) at distinct time points (d0, d2, d6, and d12).

## Time dependent cell response profiling

We used a real time cell analysis system, xCELLigence RT-CA (Roche) to monitor motility of hMSCs. The system detects impedence changes within multiwells that contain electrode arrays at the bottom (E-plates). As the culture cells multiply, this increases impedence and an upward deflection is recorded in the *cell index*. Therefore, proliferating and migrating hMSCs show an increasing cell index, while neuronal differentiated hMSCs are expected to display stable cell index related to their decreased motility. Before starting the experiments background readings were made taken with culture medium only. Then, $3x10^3$ hMSCs were added to each well in expansion medium, which was replaced with the neuronal induction medium 24 hours later. The induction medium was replaced every 48 hours. The E-Plates containing the cells were placed on the reader in 37˚C, 5% $CO_2$ incubator for cell index recording for 12 days.

## Cell viability assays

For analysis of cell death on neuronal induced hMSCs, cells were seeded on 96 well plates for Annexin-V and Sytox green stainings. Cells were treated with NI media and apoptosis was induced in positive control cells by treatment with camptothecin (Sigma). The medium was removed from the wells, cells were fixed with 4%PFA and 100 μl of Annexin-V-Alexa 568 labeling solution (Roche) and 50 μM Sytox Green dye (Invitrogen) was added to each well. After 10 or 15 minutes incubation at 15 to 25˚C, wells were washed with PBS. As for nuclear staining, cells were treated with 1:15000 DAPI (Sigma) for 10 minutes at RT. Wells were washed with PBS and dH$_2$O subsequently. The cells were analyzed by fluorescence microscopy (Zeiss Inverted Microscope with Hoffman Modulation).

## Reverse transcriptase PCR (RT-PCR)

RNA samples of hMSC were extracted by using RNeasy kit (Qiagen). 0.5 μg of total RNA was reverse transcribed to obtain cDNA by Quantitect Reverse Transcription kit (Qiagen). cDNA library was obtained after 35 cycles of amplification (PCR core kit, Qiagen). Primer pairs (Forward; Fw and reverse; Rv) used in the experiments as follows: βIII tubulin (Fw: 5'–AGTGATGAGCATGGCATCGA–3' and Rv: 5'–AGGCAGTCGCAGTTTTCACA–3') generating a 317 bp fragment; NSE (Fw: 5'–CCCACTGATCCTTCCCGATACAT–3' and Rv: 5'–CCGATCTGG TTGACCTTGAGCA–3') generating a 254 bp fragment; NF-L (Fw: 5'–TCCTACTACACCAGCCATGT–3' and Rv: 5'–TCCCCAGCACCTTCAACTTT–3') generating a 284 bp fragment; Nestin (Fw: 5'–TGGCTCAGAGGAAGAGTCTGA–3' and Rv: 5'–TCCCCCATTTACATGCTGTGA–3') generating a 148 bp fragment. A human GAPDH primer pair (Fw: 5'–GTCAGTGGTGGACCTGACCT–3', Rv: 5'–TGCTGTAGCCAAATTCGTTG–3') generating a 245 bp fragment was used as a positive control.

## Immunofluorescent staining

hMd-Neurons were analysed for neuron specific phenotypical characteristics at day 12 of neuronal induction. Each staining was performed as triplicated wells and repeated for each donor. Cells were fixed with 4% paraformaldehyde in phosphate buffer saline (PBS, Sigma) and incubated at RT for 15 minutes. After washing with PBS, cells were blocked with the solution containing 1% Goat serum (Sigma), 3% BSA (Sigma), 0.3% Sodium azide (Sigma), and 0.1% Triton X-100 in PBS. After discarding blocking solution, cells were treated with primary antibodies against neuron specific enolase (NSE (2.5 μg/ml diluted), Abcam; ab53025), neuronal specific nuclear protein (NeuN (1:500 diluted), Abcam; ab104224), protein gene product 9.5 (PGP9.5 (1:100 diluted), Abcam-ab8189), microtubule associated protein 2 (Map2 (1:500 diluted), Sigma; M1406), neurofilament 200 (NF200 (1:200 diluted), Abcam; ab4680), Synaptophysin ((1:500 diluted) Abcam; 17785-1-AP) and PSD-95 ((2.5 μg/ml diluted) Abcam; ab12093) using desired concentrations in PBS containing 3% BSA (Sigma), 0.3% Sodium azide (Sigma), %1 Tween20 and 1% Goat serum (Sigma) at 4˚C for o/n. After antibody treatment, wells were washed with PBS and incubated at room temperature for three hour with 1:200 diluted goat anti-mouse (GAM) IgG Alexa Fluor 488 (Abcam; ab150113), 1:200 diluted goat anti-rabbit (GAR) IgG Alexa Fluor 488 (Abcam; ab150077), 1:200 diluted donkey anti-goat (DAG) IgG Alexa Fluor 568 (Abcam; ab175474), 1:500 diluted goat anti-rabbit (GAR) IgG Alexa Fluor 594 (Abcam; ab150088) and 1:100 diluted goat anti-chicken (GAC) IgG Alexa Fluor 633 (A-21103) secondary antibodies. After washing with PBS, cells were treated with 1:15000 DAPI solution and incubated at RT for 3 minutes. Wells were washed with PBS containing 0.1% sodium azide and slides were mounted with vectashield mounting medium (Vector Lab). Cell images were taken with fluorescent microscope (Zeiss LSM780 Confocal Microscope).

## Western blotting

For western blot sample collection hMSCs were seeded at $1.0–1.5 \times 10^5$ cells/well into glass bottom 6 well plates and were incubated at 37˚ C in 5% $CO_2$ until the next day. Neuronal inducton protocol was applied as described above. Then protein lysates at distinct days (d) of neuronal differentiation (d0, d2, d6, and d12) of hMd-Neurons from hMSC line were obtained using Ripa lysis buffer (Thermo Fischer Scientific; #89900). Equal amounts of protein samples were run on SDS-PAGE and iBlot-2 Dry Blotting (Thermo Fisher Scientific) system was used according to manufacturer's recommendations. Then the membranes were probed with following antibodies; anti-βIII Tubulin (Abcam #ab18207) (1:1000), anti-NSE (Abcam #ab53025) (1:750) and anti- β actin (Santa Cruz #sc-47778) (1:1000). As for secondary antibodies; anti-

rabbit IgG, HRP-linked antibody (CST; #7074) (1:2000) and goat anti-mouse IgG-HRP-linked antibody (Santa Cruz #sc-2005) (1:2000) were used. After applying ECL substrate (Bio-Rad), the protein bands were chemiluminescently detected in ChemiDoc MP Imaging System (Bio-Rad).

## Calcium imaging

To analyse changes in $Ca^{++}$ concentration in neurons from human bone marrow mesenchymal stem cells (hMd-Neurons), we used Fluo-4 staining (ThermoFisher) between day5-day18 of neuronal induction. Briefly, medium was aspirated and washed with Hank's Balanced Salt Solution (HBSS, Sigma) and Tyrode's solution. Cells were treated with staining solution containing Fluo-4, pluronic acid and Tyrode's solution at room temperature for 30 minutes. After discarding staining solution, cells were washed with Tyrode's solution and HBSS. Cells were incubated at 37˚C for 10 minutes in induction medium. Then, fast imaging was done on Zeiss Cell Observer SD Spinning Disk Time-Lapse Microscope. calcium imaging was performed for 150–250 cells for each group and repeated for different donors as well.

## Electrophysiology

We maintained hMd-Neuron culture for longer periods of time in NI media to enable progress in neuronal maturation. Before patch clamp recordings to determine the viability of neuronal induced hMSC for longer culture periods (during 21 days), we used CellTiter Glo Viability Assay (Promega) following the protocol according to manufacturer's instructions. A 15 mm coverslip containing donor derived hMd-Neurons at day 5–18 of neuronal induction was placed into the recording chamber. Cultured cells were perfused with aCSF (artificial cerebrospinal fluid) containing the following (in mM): 150 NaCl, 10 D-glucose, 4 KCl, 2 $MgCl_2$, 2 $CaCl_2$, and 10 HEPES, aerated with 95% $O_2$, 5% $CO_2$ delivered at a rate of 2–3 ml/min. The cells were identified and targeted using an Olympus microscope and were patched using pipettes with 3–6 MΩ tip resistances in the bath when filled with an internal solution. Whole cell voltage-clamp recordings were performed using pipettes (Harvard Apparatus) made from borosilicate glass capillaries pulled on a Flaming-Brown micropipette puller (Model P-1000, Sutter Instruments, Novato, CA). Pipette solution contained (in mM): 125 CsCl, 5 NaCl, 10 HEPES, 0.6 EGTA, 4 Mg-ATP, 0.3 $Na_2GTP$, 10 lidocaine N-ethyl bromide (QX-314), pH 7.35 and 290 mOsm. The holding potential was set to -60 mV. Electrical stimulation performed using a field electrode that was placed within adjacent to the coverslip. Half-maximal stimulus strength was used for electrical stimulation, we waited at least 200 milliseconds between successive field stimuli. All experiments were performed at room temperature as triplicates.

## Statistics

Statistical comparisons were performed by *t*-test and/or two-tailed *t*-test assuming equal variance. Differences were considered as statistically significant at $p^{***} < 0.0001$. Data are the mean ± standard error (SE). One-way analysis of variance (ANOVA) was used to evaluate the differences in cell death levels among three groups. An alpha value of $p^{\#} < 0.05$ was used for statistical significance.

## Results

### Functional neurons (hMd-Neurons) can be yielded with high ratios from hMSCs *in vitro*

To study neuronal differentiation of hMSCs, we initially used human bone marrow hMSC cell line (UE7T-13 cells, no. RBRC-RCB2161; RIKEN, Japan) [28–32]. To improve *in vitro*

generation of neuronal cells with sufficient differentiation capacity, we used a non-viral neuronal induction method which is an enriched form of previously defined combination [33]. Neuronal cell morphology was observed within 24 hrs upon neuronal induction (NI) and almost all hMSC cell line gave rise to bipolar neuron-like cells with neuritis (Fig 1A).

For neuron specific characteristics, we stained hMd-Neurons for neuronal markers including NF, NeuN, NSE, PGP 9.5, as well as synaptic proteins Synaptophysin, and PSD 95 on day 10 of NI. hMd-Neurons showed expressions of all neuronal markers with high percentages (Fig 1B and 1C, and S1A Fig). Moreover, both hMd-Neurons from hMSC cell line and uninduced hMSC cell line showed NSE and βIII tubulin transcripts and protein expressions (S1B and S1C Fig).

To evaluate spontaneous activity of hMd-Neurons, we performed live cell $Ca^{++}$ imaging without any chemical addition, which showed $Ca^{++}$ transients in differentiated hMSCs. More than 78% of hMd-Neurons were spontaneously active neurons showing $Ca^{++}$ concentration changes without any stimulation. They showed spontaneous activity that is not induced by an external stimulus with different firing patterns [34] (Fig 1D–1F, and S1 Video).

## Isolated cells from healthy bone marrow donors represent hMSC phenotype

After yielding functional neurons from hMSC cell line, we then studied neuronal differentiation of hMSCs from healthy human bone marrow donors in detail. We first showed that cells isolated from human bone marrow are MSCs in agreement with the criteria of International Society for Cellular Therapy published in 2006 [35]. During initial phases of culture, non-adherent cells were depleted and very small proportion of the cells attached on plastic culture surface. These cells had a fibroblast-like morphology, formed colonies and by passage 3 (P3), more than 95% of the cells were MSCs as revealed by flow cytometry analyses. Results from different healthy donors indicated that these adherent cells at P3 have hMSC immunophenotype with negative expression of CD45, CD34, CD14 and CD31 while they were positive for CD29, CD44, CD73, and CD105 (Fig 2A and 2B).

For lineage specific MSC differentiation ability, we analysed mesodermal differentiation (adipogenic, osteogenic and chondrogenic) of immunophenotyped hMSCs. To demonstrate adipogenic differentiation of hMSCs, we used Oil Red-O, which labels neutral triglycerides, lipids and lipoproteins. For detection of chondrogenic differentiation, Toluidine Blue-O and Alcian Blue stains were applied; while the former stains calcium deposits indicative of mineralization, the latter turns chondrogenic spheroids into dark blue which marks the extracellular matrix of cartilage (Fig 2C). Positive stainings were counted for each from 10 different independent fields and the differentiation percentages were then determined. Depending on 50% chondrogenic, 43% osteogenic and 52% adipogenic differentiation rates, we concluded that more than 95% of the cells from bone marrow were hMSCs at P3 in our experimental paradigm.

## Donor derived hMSCs can also differentiate into hMd-Neurons

We maintained NI for 12 days prior to phenotypical/functional characterization of neuronal induced hMSCs from bone marrow donors and their neuronal morphology in culture was variable (Fig 3A). Morphologically neuronal cells with neurite extensions were counted from 10 different independent fields and the neuronal morphology percentages were then determined. On one hand, hMSCs derived from bone marrow donors showed neuronal morphology (65%) at day 1 of NI while neuronal cell percentage was increased to 80% by day 3 (S2a and S2b Fig). We determined the highest proportion of neuronal cells from hMSC donors as 85% at day 5 (S2 Fig), which slightly increased up to 90–95% at day 12 (Fig 3B).

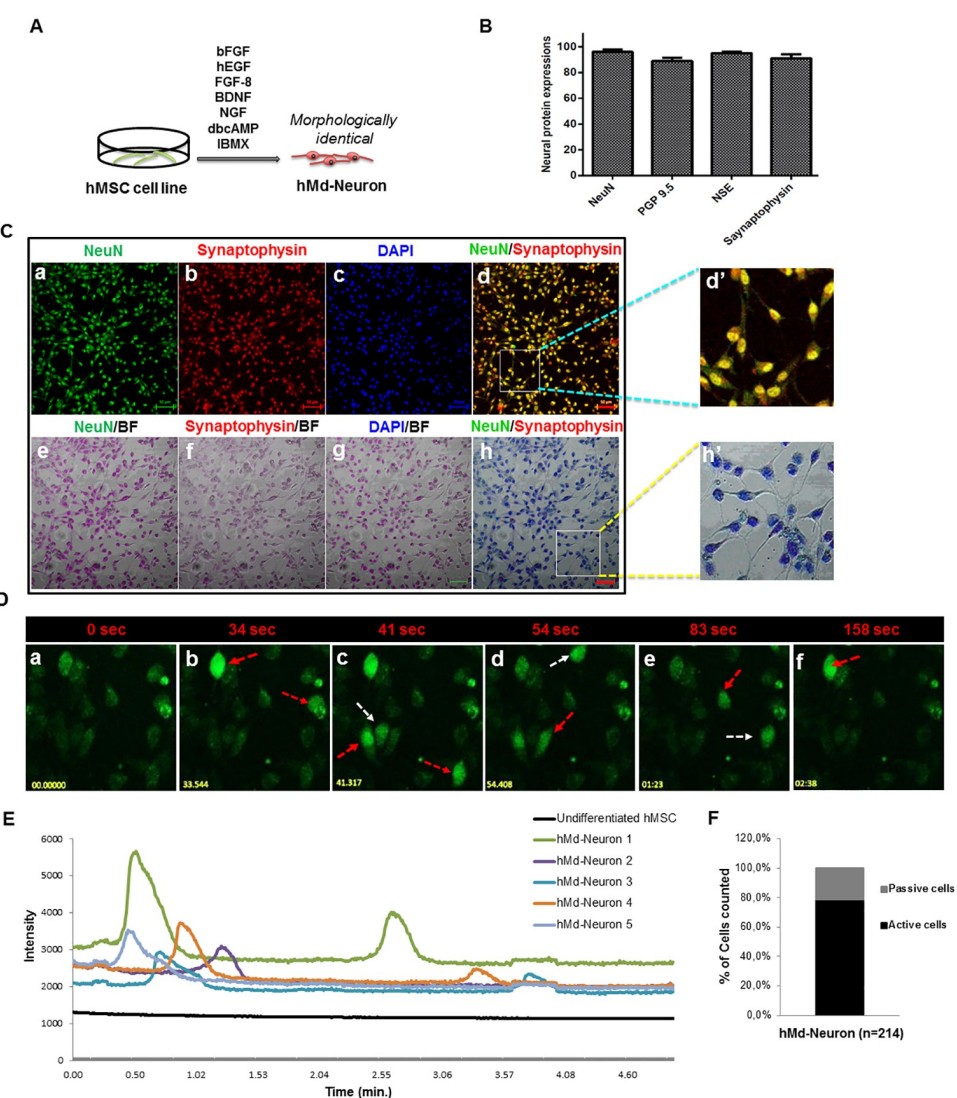

**Fig 1.** hMSC cell line from bone marrow has the ability to differentiate into spontaneously active neurons (A) Schematic representation of neuronal induction on hMSC cell line. (B) Plot indicates neuronal markers expression percentages of hMd-Neurons from hMSC cell line after neuronal induction during 12 days and almost %100 of neuronal induced cells express neuronal maturation proteins NeuN, Synaptophysin, NSE and PGP 9.5. Positively stained cells counted from 10 different area of staining and averages were calculated. Functionality of hMd-Neurons was evaluated upon labeling with Fluo-4 for real time Ca$^{++}$ ion imaging without any outside stimulation chemically. (C) Immunofluorescence co-staining of hMSC cell lines in neuronal induction medium; NIM composed of NGF, BDNF, FGF-8, bFGF, EGF, dbcAMP, IBMX, B27 for 12 days reveals the presence of neuronal maturation proteins NeuN (a, e) and Synaptophysin (b, f) with DAPI nuclear staining (c, g). Merged images represent positive co-staining of NeuN and Synaptophysin for each individual cell (d, h). Dashed yellow squares magnified 3 folds (d', h'). (D) Florescent images (a-f) demonstrates time dependent firing pattern of hMd-Neurons from hMSC cell line through imaging of Ca$^{++}$ ion kinetics and arrows indicate firing of each hMd-Neuron separately (sec; seconds). (S1 Video) (E) Histogram indicates firing frequency and signal intensity of each individual hMd-Neuron while there is no signs of spontaneous activity from uninduced hMSCs. According to Ca$^{++}$ influx/efflux through hMd-Neurons, (F) 78.5% of cells were recorded as spontaneously active with twice firing frequency in 4 minutes. Data are represented as mean ± S. E.M. Scale bars represent 50 μm.

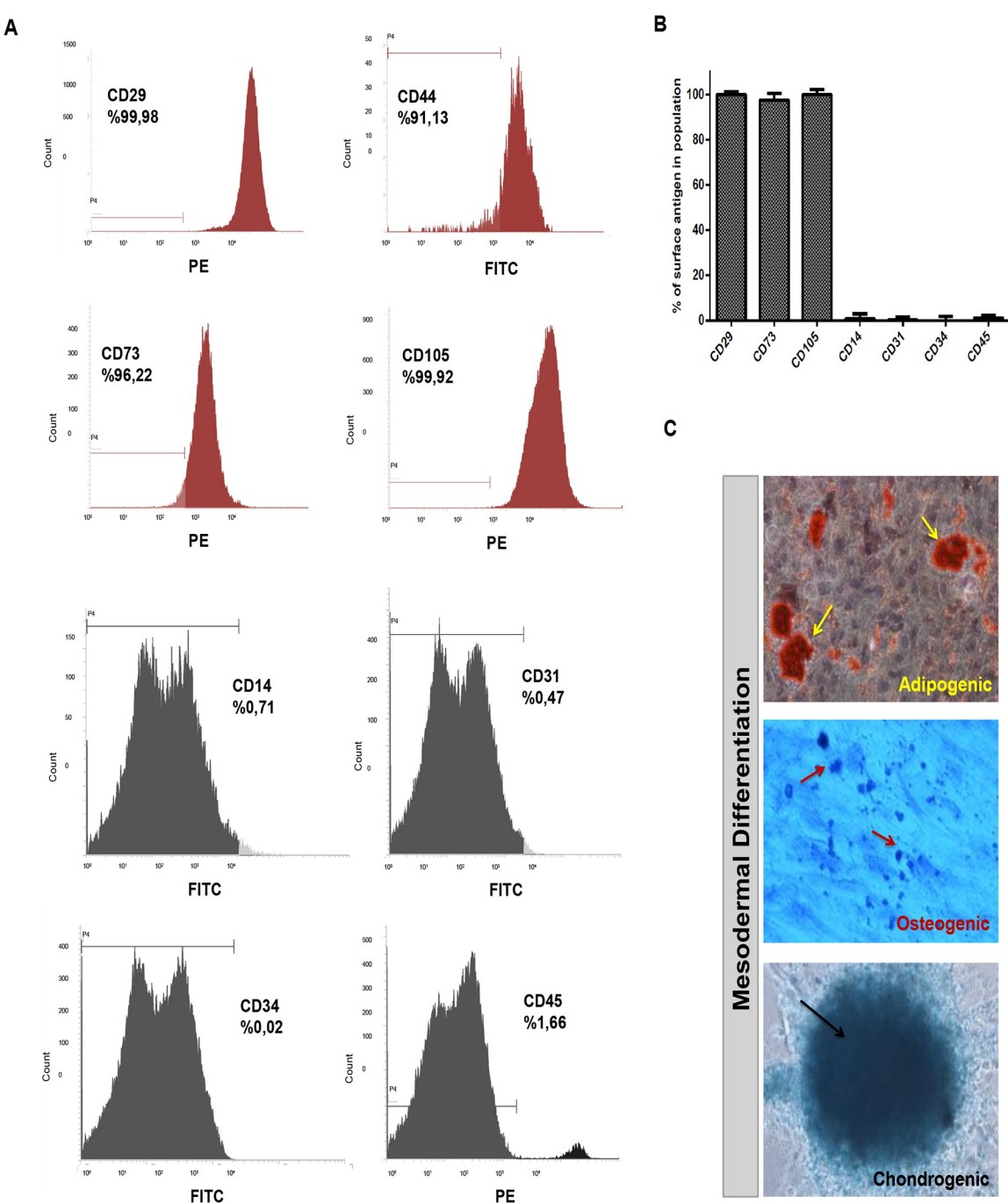

**Fig 2.** Isolated cells from bone marrow stroma of healthy donors display typical hMSC phenotype and mesodermal differentiation capacity (A, B) After isolation of human bone marrow cells from healthy donors, cells at passage 3 were analyzed by flow cytometry for phenotypic characterization. The cells showed hMSC specific marker expressions with negative expression of CD14, CD31, CD34, and CD45, while they were positive for CD29, CD44, CD73 and CD105. Representative histograms and surface antigen profiles of isolated adherent cells from 5 healthy donors are demonstrated. (C) Adipogenic differentiation was confirmed by Oil Red-O staining of lipid vacuoles (with hematoxylin counterstain); osteogenic differentiation was followed with Toluidine Blue staining of calcium deposits showing mineralization; and Alcian Blue staining of proteoglycans in chondrogenic pellet demonstrated chondrogenic differentiation.

Prior to further analysis on hMd-Neurons from bone marrow donors, we determined the toxicity of neuronal induction media (NI). For this, we evaluated cell death and apoptosis during neuronal induction of hMSCs from bone marrow donors at day2, day6, and day12. An apoptosis inducer camptotechin (CAM) was used as a positive control. Annexin-V and Sytox Green stainings showed that while 89±11% of CAM-treated cells were dead at day 2, neuronal induced hMSC were viable with a high rate comparable to uninduced control cells. At day 6, we observed 90±6%, 10±3%, and 4.5±2% of dead cells in CAM-treated, uninduced and neuronal induced cells, respectively. At day 12, 94±4%, 20±3% and 10±2% of cells were dead in CAM treated, uninduced, and neuronal induced cells, respectively. Of all dead cells 99±2%, 99±2% and 99.5±1% of hMSCs were apoptotic in CAM group at day 2, 6 and 12, respectively. In the uninduced group 1±0.8% and 15±2% apoptotic cells were observed only at day 6 and day 12. In contrast, a no significant number of apoptotic cells were observed in NI group at any time point (Fig 3C and 3D). Next, we investigated cell kinetics of differentiated versus undifferentiated hMSCs by real time cell analysis depending on cel motility changes (named as cell index) (XCelligence, Roche). The recordings suggest that the number of uninduced hMSCs incresad steadily while hMd-Neurons showed a stable cell index by day2 of NI. In addition to that, cell index was stabilized around day 6 of NI at which cells might commit to attain neuronal characteristics (Fig 3E).

As a phenotypical outcome of neuronal differentiation, we assessed neuronal transcripts of βIII Tubulin, Nestin, NSE, NF, and a housekeeping gene (*GAPDH*). RT-PCR results indicated that donor derived hMd-Neurons have transcripts of early neuronal marker NSE by day 2 and late neuronal marker NF by day 10. Transcripts of βIII Tubulin and Nestin were detected during all stages of differentiation. However, uninduced hMSC also had βIII Tubulin and Nestin mRNAs (Fig 3F), which has also been previously reported [36–39]. Next, we showed neuronal protein expressions on donor-derived hMd-Neurons on day 12 of neuronal induction (NI) by immunofluorescence staining. More than 90% of hMd-Neurons showed expressions of neuronal markers NF, NeuN, NSE, PGP 9.5, as well as synaptic proteins Synaptophysin, and PSD 95 (Fig 3G and 3H).

## hMSCs preserve Nestin whereas they loose Ki-67 and Sox-2 expression during differentiation into hMd-Neurons

To examine the differentiation process of hMd-Neurons, we evaluated the fractions of Nestin and Sox-2 positive cells in hMd-Neurons at day 6. We also included Ki-67 as a marker for proliferating cells. Flow cytometry analysis (single channel) showed that among hMd-Neurons from healthy donors 99% were Nestin positive; 96% of cells were Ki-67 and Sox-2 negative (Fig 3I). Accordingly, multichannel records revealed that almost 90% of Nestin positive cells were Sox-2 and Ki-67 negative whilst Nestin positive (Fig 3J). This data correlated with the presence of Nestin transcripts during neuronal differentiation of donor derived hMSCs (Fig 3F). Assembly of data from mature protein expressions and lack of stem cell and proliferation markers suggests that phenotype of hMd-Neurons by day 6 is a differentiated neuronal cell type rather than a neuronal stem/progenitor cell type.

## hMd-Neurons from healthy bone marrow donors mature into both electrophysiologically and spontaneously active neurons

To study functionality of hMd-Neurons from health bone marrow donors, we stained their *in vitro* neuronal connections for presynaptic (Synaptophysin) and postsynaptic (PSD95) proteins and they were positive for both synaptic markers (Fig 4A).

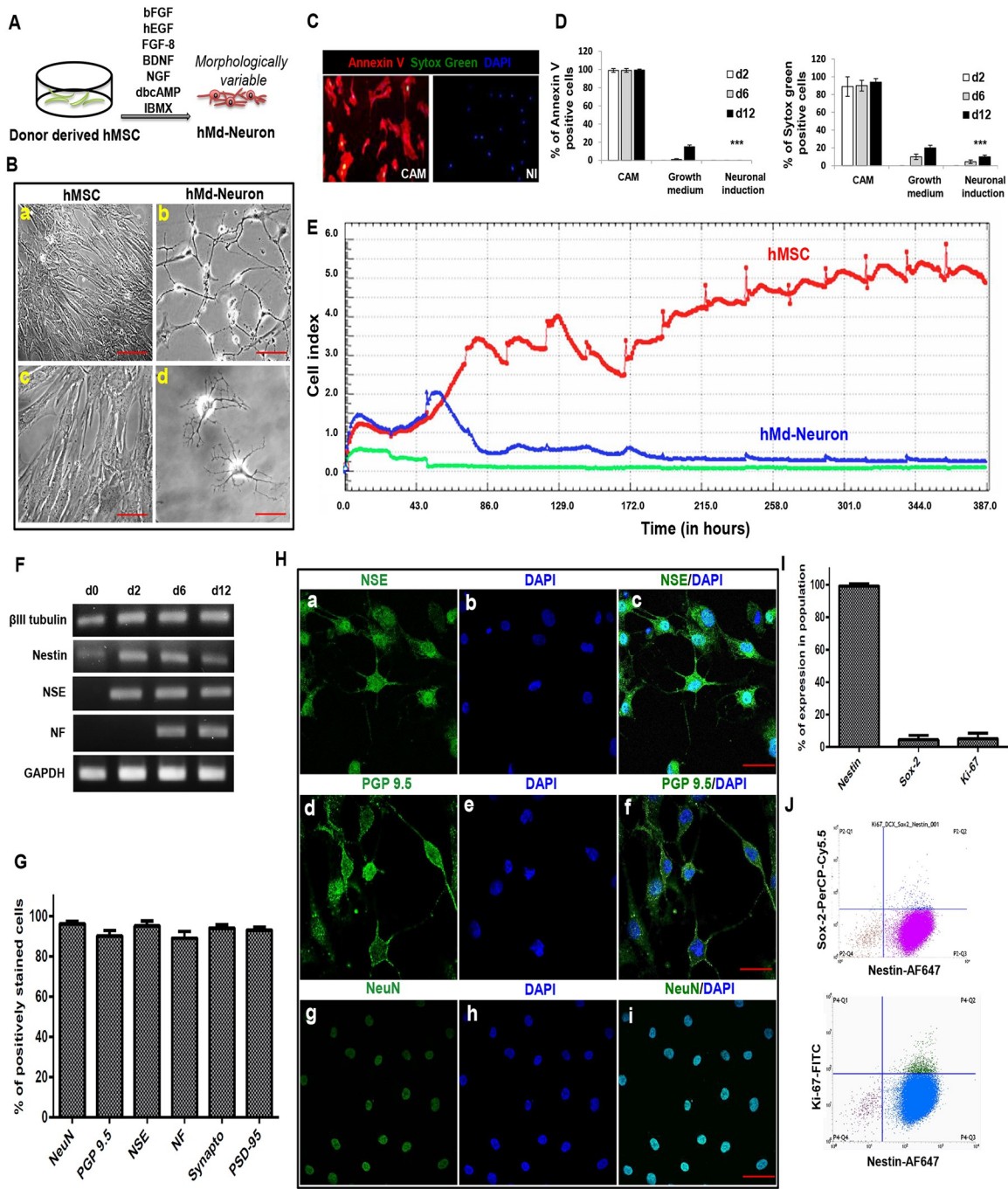

**Fig 3.** Bone marrow hMSCs from healthy donors differentiated into hMd-Neurons with phenotypical characteristics (A) Schematic representation of neuronal induction on bone marrow derived hMSCs from healthy donors. (B) Bright field images represents almost % 90 of neuronal induced human bone marrow donor derived hMSCs give rise morphologically variable neuron like cells on culture at day 12 (b, d) whilst uninduced hMSCs maintains proliferation (a, d). Morphology of hMd-Neurons from human samples varies in culture as seen with neurite extension patterns (b, d). (S2 Fig). (C) Representative florescent images of Annexin V/Sytox green stainings showing cell viability of neuronal induced hMSCs with positive control stainings of dead cells on camptothecin (CAM) treated uninduced hMSCs. (D) Positively stained cells from Annexin V and sytox green stainings of 3 different donors were counted and the percentages of dead cells were determined. Plots indicate dead cell percentages from d2 to d12 in CAM treated, untreated (in growth medium), and neuronal induced hMSCs. (E) Graph showing time dependent cell proliferation profile of uninduced and neuronal induced hMSC during 12 days in culture by X-Celligence (Roche) real time cell analysis (RT-CA) system (F) RT-PCR demonstrating presence of βIII tubulin, Nestin, NSE, NF transcripts during neuronal differentiation (d2-d12) respectively versus an absence in uninduced hMSCs (d0). (G) Plots indicating neuronal marker expression percentages ≥%89 (Images from 3 different donors were counted to determine the

neuronal marker percentages). Positively immunostained cells counted from 10 different areas of stainings and averages were calculated. (H) Representative confocal images showing neuronal protein expressions; NSE, PGP 9.5 and NeuN respectively (a, d, g) with DAPI nuclear stainings (b, e, h) of hMd-Neurons at d12 after neuronal induction. Fluorescent images of neuronal protein expressions merged with DAPI staining (c, f, i). (I) Plots represents flow cytometry results indicating 99.24% of hMd-Neurons are Nestin positive and negative for Ki-67 proliferation marker and Sox-2 expressions at day6of NI. (J) Flow cytometry analysis indicates loss of Ki-67 proliferation marker and Sox-2 expressions in Nestin positive cells. Scale bars represent 100 μm. Data are represented as mean ± S.E.M. Significance of ANOVA test *** p <0.0001.

We then evaluated their $Ca^{++}$ ion flux as we demonstrated for hMSC cell line derived hMd-Neurons. We recorded spontaneous activity of donor-derived hMd-Neurons through $Ca^{++}$ ion imaging (Fig 4B, S3 Fig and S2 and S3 Videos). No activity was detected from uninduced hMSCs (S4 Video). Donor derived hMd-Neurons showed individual differences in terms of

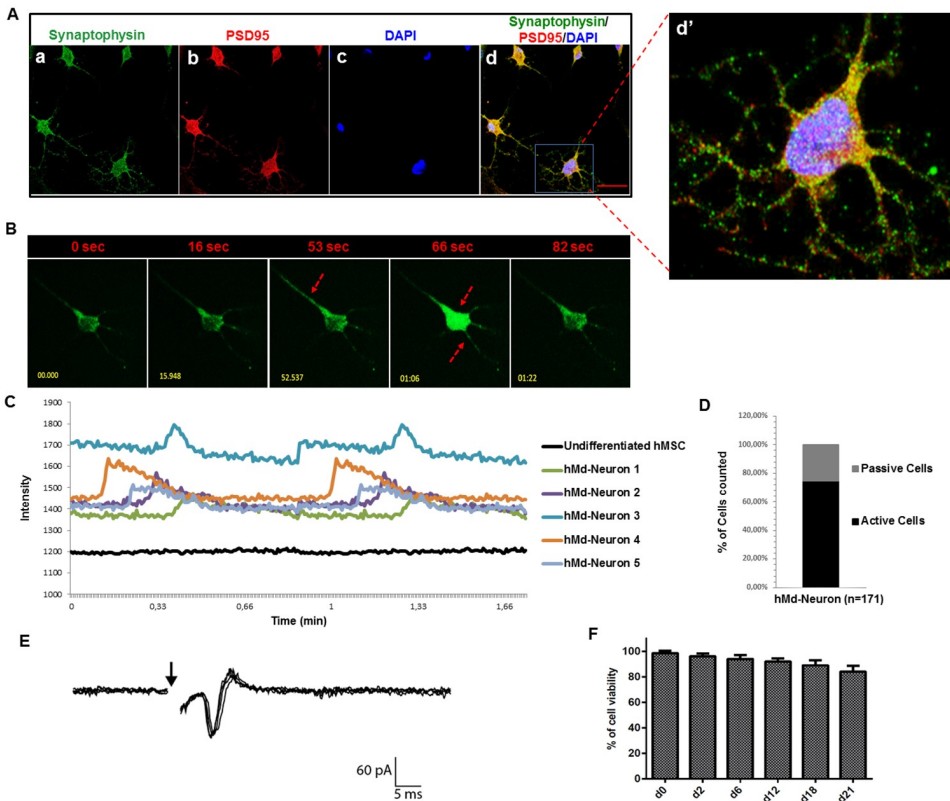

**Fig 4.** Donor derived hMd-Neurons mature into functional neurons with both spontaneous and electrophysiological activity Confocal images of neuronal induced hMSCs from healthy bone marrow donors (2–10 aged) at day 12 reveals the expression of synaptic proteins (a) Synaptophysin and (b) PSD-95 with (c) DAPI nuclear stainings. Merged image represents positive co-staining of Synaptophysin and PSD-95 for each individual hMd-Neurons (d). Dashed yellow squares magnified 5 folds (d'). Functionality of hMd-Neurons was evaluated using Fluo-4 calcium indicator for real time $Ca^{++}$ ion imaging without any stimulatory chemical addition. (B) Florescent images (e-i) represents real time firing patterns of hMd-Neurons from hMSCs via imaging of $Ca^{++}$ signal transmission through a single neuron and arrows indicate initiation and termination of firing in seconds (sec). (S3 Fig and S2 Video). (C) Representative histogram indicates firing frequency of each individual hMd-Neurons as twice in less than 1 minute. No signs of activity of uninduced hMSCs of the same donors were detected. (D) Upon imaging $Ca^{++}$ ion dynamics, 74,6% of cells were counted as spontaneously active according to firing situation (percentages were determined from average of 3 different donors). (E) RepresANTative patch clamp (whole cell) recordings of donor derived hMd-Neurons upon 18 days neuronal induction. (F) Prior to electrophysiology experiments, long term hMd-Neurons viability was evaluated for further maturation in culture using Cell Titer Glo (Promega) assay. Data are represented as mean ± S.E.M. Scale bars represent 50 μm.

firing frequencies (average was twice in less than 1 minute) (Fig 4C). Among all hMd-Neurons, around %75 were spontaneously active neurons (Fig 4D).

In addition to Ca$^{++}$ imaging, we evaluated neuronal functionality by patch clamp technique. Prior to experiments we cultured hMd-Neurons for 18 days to let maturation of their electrical properties. We also recorded directly from donor-derived hMd-Neurons to characterize their electrophysiological properties. We applied electrical stimulation to activate the cells and then hMd-Neurons generated action potentials upon electrical stimulation in miliseconds similar to a maturating neuron (Fig 4E). We also showed that hMd-Neurons could survive as monolayers over long periods in NI media without the addition of any extracellular matrix proteins or coating materials (Fig 4F).

## Discussions and conclusion

One of the main concerns about studies with neuronal differentiation of hMSCs is establishing a successful induction method leading to sustainable and high yields of functional neurons [40]. In this study, we showed that bone marrow hMSCs either from cell line or healthy donors differentiate into functional neurons (>74%) by a single step protocol. hMSC derived neurons (hMd-Neurons) displayed spontaneous activity and showed response in miliseconds to electrical stimulation as a typical maturating neuron [41].

We first focused on the phenotype of neuronal induced hMSC cell line and donor derived hMSCs prior to functionality analysis. Immunostainings showed that induced hMSCs in the presence of EGF, bFGF, NGF, BDNF, FGF-8, dbcAMP and IBMX are able to progress into neuronal differentiation with positive expressions of neuronal proteins (Nestin, NeuN, NF, Synaptophysin, PSD95, PGP9.5). Moreover, we detected Nestin expression during 12 days of neuronal differentiation whereas proliferating cells had depleted Nestin expressions. Accordingly, our cell fractionation analysis for neuronal induced hMSCs showed that almost 90% of Nestin positive cells were also negative for Ki-67 and Sox-2 indicating that they were in the process of neuronal maturation. Additionally, real time cell response of neuronal induced hMSCs also supported that on the way to differention into hMd-Neurons, proliferation modalities were downregulated which associates with post-mitotic nature of mature neurons. Existance of neuronal phenotype of hMd-Neurons was also seen in neurite outgrowths which is also crucial for the establishment and maintanance of neuronal networks [42]. hMd-Neurons exhibit neurite to neurite and neurite to cell body end points which evokes axoaxonic/axodentric and axosomatic synapses. First set of experiments remarkably revealed the dominant neuronal characteristics of induced hMSCs from both cell line and bone marrow donors, then ultimately we focused on inquiries about neuronal functionality and activity ratio of hMd-Neurons.

Functional connections on hMd-Neurons were initially profiled for the expression of presynaptic (Synaptophysin) and post-synaptic (PSD95) proteins as synaptic formation is required for communication of neurons in their own network. hMd-Neurons from both hMSC cell line and primary hMSCs visibly built synaptic connections *in vitro*. We then monitored calcium ion exchange to determine activity of hMd-Neurons. Surprisingly, we could detect spontaneous activity of hMd-Neurons both from hMSC cell line and donor derived hMSCs without any stimulatory chemical addition [21,43,44]. This is a phenomenon that can be discussed since spontaneous activity is important for neuronal specification during neurogenesis but not an ability of all primary neurons. For instance, when calcium ion imaging applied, mouse derived dorsal root ganglia (DRG) neurons are quiescent (no activity) while cortical neurons display firing pattern spontaneously even at high frequencies [45,46]. Therefore further investigations are required on hMd-Neurons (or other functional neurons derived

from stem cells) to well define their capabilities, specifications and tissue/function specific phenotypes.

Another outcome of calcium imaging experiments showed that timing of $Ca^{++}$ ion flux of hMd-Neurons was varying. Mainly, spontaneous activity of hMd-Neurons was recorded at day 5. On the other hand, hMSCs from one of the donors were able to function prior to day 5 of NI. What is more, hMSCs from one of the donors displayed spontaneous activity around day 8 while no signals were recorded at day 5. Regardless of difference in activation time point, all of the donors sustained the functionality during their *in vitro* neuronal maturation. Overall, hMd-Neurons mostly showed activity at day 5, our data pointed out individual differences of hMSC sources in terms of neuronal functionality timing.

Following the neuronal activity records, we further investigated the excitability of hMd-Neurons from 5 different human bone marrow donors to test their synaptic functionality. According to patch clamp results hMd-Neurons were electrophysiologically active and showed a similar response pattern of a neuron on maturation [47]. Additionally, improvement of the culture conditions using tissue engineering tools, which mimicks neuronal microenvironment can be applied to study maturation process of hMd-Neurons thoroughly. These studies will maximize to understand abilities of hMd-Neurons to form functional neuronal networks and this is an avenue that we are actively pursuing.

In terms of comparing hMd-Neurons from MSC cell line and primary hMSCs, we observed that hMd-Neurons from hMSC cell lines were identical and mostly had bipolar structures. Otherwise, hMd-Neurons from human bone marrow donors displayed heterogenity in neurite extension patterns. This explains that since cells of hMSC cell line are genetically identical, they give rise to less complex neurons with structural similarities. Moreover, hMSC cell line used in this study shows several neuronal marker expressions without inducing into neurons while donor derived hMSCs showed absence or low expressions of neuronal proteins. Actually, hMSCs are expected to express several proteins as they are stem cells with the ability of differentiation. Additionally, since the expression of proteins do not indicate their functional roles, it is better to investigate whether they establish synaptic networks or release measurable calcium currents. Eventhough hMSC cell line express neuronal markers, our $Ca^{++}$ imaging results showed that they can only form functional connections when they are neuronally induced. Many studies using neural induction protocols only analyse the expression pattern of neuron spesific proteins and progress into differentiation mechanisms of hMSCs. These findings support that researchers might focus on not only the expression of synaptic neuronal markers, but also have to investigate functionality of those proteins before further analysis of mechanism.

Conversely, primary hMSCs from bone marrow donors might have different subpopulations with seperate differentiation abilities. Therefore immunophenotyping of isolated cells from bone marrow as hMSCs only allows researchers to classify hMSCs for general characteristics and not for their differentiation capacity leading to specific type of a neuron. Furthermore, hMd-Neurons can be served as a potential candidate to highly yield certain types of neurons such as Dopaminergic, Gabaergic neurons etc. as many researchers focused on this field for therapeutic applications [48–50]. One solution to that might be investigation of hMSC subpopulations that have the potential to differentiate into a specific neuron type with high purity under certain conditions. Similar approach was previously reported to successfully yield neurons from Nestin positive hMSCs [51,52]. Adopting it to obtain function/tissue specific neurons from hMSC subtypes can be a clonogenic based approach for neuronal differentiation studies. We are in the process of derivation of functional hMd-Neurons according to hMSC subtype. Furthermore, categorization of hMd-Neurons according to their structural class, homing certain types of neurotransmitter receptors, and overall investigation of their

function/tissue spesific abilities can pioneer the regenerative studies for a broad range of the nervous system deficiencies and ultimately lead to development of disease models and preclinical therapeutic approaches [53,54].

Taken together, our findings introduce functional hMd-Neurons, that can be easily obtained *in vitro* either from hMSC cell line or donor derived hMSCs through a defined induction protocol. According to these outcomes, in vitro characterization of hMd-Neurons for specific neuronal subtypes; hMd-Neuron derivation from patients with neuropathologies and their *in vivo* functional abilities can be further evaluated. Therefore, hMd-Neurons can be utilized for a wide range of applications including, disease modeling; tissue engineering approaches; studying the mechanism of neuronal differentiation; and understanding hierarchic establishment of newborn neurons from hMSCs.

## Supporting information

**S1 Video. Real time calcium imaging of hMd-Neurons from hMSC cell lines.**
(M4V)

**S2 Video. Single cell focusing of calcium imaging in hMd-Neurons from bone marrow donor derived hMSCs.**
(M4V)

**S3 Video. Real time calcium imaging of hMd-Neurons from bone marrow donor derived hMSCs.**
(M4V)

**S4 Video. Real time calcium imaging in uninduced hMSCs.**
(M4V)

**S1 Fig.** Neuronal marker expressions of hMd-Neurons from hMSC cell lines (A) hMSC cell line was stained for distinct neuronal protein expressions and almost %100 of neuronal induced hMSCs were identically positive for PGP 9.5 (a) and NSE (d) with DAPI nuclear stain. Merged images of PGP 9.5 and NSE (c, f) magnified 4 folds respectively (c', f'). Scale bars represent 50 μm. (B) RT-PCR and Western blot analysis (C) of NSE and βIII tubulin in neuronal induced hMSC cell line during 12 days.
(TIF)

**S2 Fig.** Neuronal cell morphology with neurite extensions appears by day 1 of hMSC neuronal induction (A) Bright field images represent morphology of hMd-Neurons from healthy bone marrow donors in culture by d1, d3 and d5 (a, b, c). Images were taken under 10X. Dashed squares magnified 2 folds respectively (a', b', c'). Arrows indicate neurite to neurite and neurite to cell body end points.
(TIF)

**S3 Fig.** Real time firing pattern of hMd-Neurons from donor derived hMSCs in a group of cells within 90 seconds (A) Florescent images (a-e) demonstrates time dependent firing pattern of hMd-Neurons from donor derived bone hMSC through imaging of $Ca^{++}$ ion influx/efflux. Numbers indicate firstly tracked signal input (1–3) and output (1' and 2') in images for some of the hMd-Neurons separately. Images were taken under 20X.
(TIF)

**S1 Data.**
(RAR)

## Acknowledgments

We thank to Deniz Atasoy for his contributions in patch clamp data records and analysis. We also thank to Emre Vatandaşlar for flow cytometry data processing.

## Author Contributions

**Conceptualization:** Nihal Karakaş, Khalid Shah, Fikrettin Şahin, Gürkan Öztürk.

**Data curation:** Nihal Karakaş, Sadık Bay, Nezaket Türkel, Nurşah Öztunç, Merve Öncül, Hülya Bilgen.

**Formal analysis:** Nurşah Öztunç.

**Investigation:** Nihal Karakaş, Sadık Bay, Nezaket Türkel, Nurşah Öztunç, Hülya Bilgen.

**Methodology:** Nihal Karakaş, Sadık Bay, Nezaket Türkel, Merve Öncül, Hülya Bilgen.

**Supervision:** Nihal Karakaş, Khalid Shah, Fikrettin Şahin, Gürkan Öztürk.

**Validation:** Nihal Karakaş.

**Writing – original draft:** Nihal Karakaş, Sadık Bay, Nezaket Türkel, Nurşah Öztunç, Merve Öncül, Hülya Bilgen.

**Writing – review & editing:** Nihal Karakaş, Khalid Shah, Fikrettin Şahin, Gürkan Öztürk.

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
