## [Decision Letter · Decision Letter 0]

28 Jan 2020

PONE-D-19-34566

Neurons from human mesenchymal stem cells display both spontaneous and stimuli responsive activity

PLOS ONE

Dear Dr. Karakas,

Thank you for submitting your manuscript to PLOS ONE. After careful consideration, we feel that it has merit but does not fully meet PLOS ONE’s publication criteria as it currently stands. Therefore, we invite you to submit a revised version of the manuscript that addresses the points raised during the review process.

The paper has been carefully evaluated by two experts in the field and both found the study of interest but requiring several amendments as fully specified in their reports. Therefore the Authors are required to revise in deep the study and manuscript and send the revised version highlighting all the alterations. 

We would appreciate receiving your revised manuscript by Mar 13 2020 11:59PM. To enhance the reproducibility of your results, we recommend that if applicable you deposit your laboratory protocols in protocols.io, where a protocol can be assigned its own identifier (DOI) such that it can be cited independently in the future. For instructions see: http://journals.plos.org/plosone/s/submission-guidelines#loc-laboratory-protocols

We look forward to receiving your revised manuscript.

Kind regards,

Gianpaolo Papaccio, M.D., Ph.D.

Academic Editor

PLOS ONE

Journal Requirements:

2. In your data availability statement you write, "All relevant data are within the paper and its Supporting Information files."

Please ensure you have provided the individual data points used to create the figures and determine means, medians and variance measures presented in the results, tables and figures (http://journals.plos.org/plosone/s/data-availability#loc-faqs-for-data-policy). If these data cannot be publicly deposited or included in the supporting information, e.g. due to patient privacy or ownership by a third party, explain why and explain how researchers may access them.

'This work was funded by Istanbul Medipol University, Regenerative and Restorative Medicine Research Center (REMER) and Yeditepe University, Department of Genetics and Bioengineering.'

'NO'

Please provide an amended Funding Statement that declares *all* the funding or sources of support received during this specific study (whether external or internal to your organization) as detailed online in our guide for authors at http://journals.plos.org/plosone/s/submit-nowPlease state what role the funders took in the study.  If any authors received a salary from any of your funders, please state which authors and which funder. If the funders had no role, please state: "The funders had no role in study design, data collection and analysis, decision to publish, or preparation of the manuscript."

5. Please upload a new copy of Figure 3 as the detail is not clear.

Please follow the link for more information: http://blogs.PLOS.org/everyone/2011/05/10/how-to-check-your-manuscript-image-quality-in-editorial-manager/

Reviewers' comments:

Reviewer's Responses to Questions

**Comments to the Author**

1. Is the manuscript technically sound, and do the data support the conclusions?

Reviewer #1: Yes

Reviewer #2: Partly

2. Has the statistical analysis been performed appropriately and rigorously? 

Reviewer #1: Yes

Reviewer #2: Yes

3. Have the authors made all data underlying the findings in their manuscript fully available?

Reviewer #1: Yes

Reviewer #2: Yes

4. Is the manuscript presented in an intelligible fashion and written in standard English?

Reviewer #1: Yes

Reviewer #2: Yes

5. Review Comments to the Author

Reviewer #1: In this paper authors showed that bone marrow hMSCs either from cell line or healthy donors differentiate into functional neurons by treating them with induction medium.

The study is interesting but the paper needs some changes.

In figures 1C, 3H and 4A immunofluorescence is not clear: authors should present fluorescence of the single proteins and DAPI in color (not only the merged panels).

Authors should better organize the paragraphs and the figure legends. In my opinion, they should put the figure legends at the end of the paragraph or of the paper.

Authors should evaluate the expression of proteins (βIII tub, Nestin, NSE. NF) at the same time points of PCR through WB analysis.

Reviewer #2: The study aims to clarify a difficult issue in the research on stem cells related to cell-based therapy/neurological disease modelling, namely the induction and maintenance of a neuron-like phenotype and neuronal functions. In the study, both cells from a bone marrow-derived human mesenchymal stem cell line and bone marrow-derived mesenchymal stem cells isolated from human donors, submitted to the established ‘single step cytokine-based induction protocol’, are demonstrated to be able to display a neuronal immunophenotype and functional activities of differentiating neurons. The Methodological and ‘Results’ sections are correct, the experimental protocols are described in detail and the results are well described and organized. However, the two main points of the study i.e. the newly-designed protocol and the demonstration of neuron-like functions, are not adequately introduced and discussed.

Major Compulsory Revisions

The Abstract, Introduction, and Discussion are incomplete or partly unclear. As an example:

Introduction - The Introduction describes the aims of the work in an incomplete fashion. It does not adequately introduce the previous advances in the field. Moreover, if it is true that dozens of compounds have been described, and in various combinations, to achieve phenotypical traces of neurons, in this manuscript the style adopted for introducing these attempts is confusing. Please rewrite.

The same criticism can be applied to the comments about the comprehension of the induction underlying mechanisms.

At the end, although several data have been reported, the more recent and specific studies and methods in the field are not highlighted. Please completely rewrite.

Discussion – A sufficient discussion is followed by a clear conclusion. Nevertheless, this reviewer has one substantial point of criticism and would like to encourage the Authors to consider this aspect in order to strengthen the entire study. Because the ability of MSCs to differentiate into cells of the neuronal lineage has been largely questioned, a more detailed, point-by-point comparison between the more recent researches, that demonstrated a successful and stable neuronal differentiation (also from a functional point of view), and the advantages suggested in the study, should be introduced.

Minor Essential Revisions

Abstract - At the beginning of the abstract, the meaning of the sentence ‘Mesenchymal stem cells are one of the promising tissue specific stem cell source’, is not clear. Please clarify. In addition, the definition ‘neural protein expressing cells’ only appears in the abstract, and is not used in the text, even when appropriate, and the same can be said for ‘neuron-like cells’ that should be used in the correct context. With the same objective, standard acronyms need to be uniformly and appropriately used throughout the text.

Please carefully control for incorrect statements, inaccuracies, and typing errors throughout the manuscript.

6. PLOS authors have the option to publish the peer review history of their article (what does this mean?). If published, this will include your full peer review and any attached files.

Reviewer #1: No

Reviewer #2: No

---

## [Author Response · Author response to Decision Letter 0]

20 Mar 2020

Dear Dr. Gianpaolo Papaccio ,

We would like to thank you for the for the helpful comments as well as valuable suggestions and the opportunity to resubmit our manuscript entitled as “Neurons from human mesenchymal stem cells display both spontaneous and stimuli responsive activity”.

Based on your suggestions, we have performed experiments, rewrote the required fields and corrected typing errors accordingly. Finally, based on your suggestions, we have edited the manuscript text, updated references and provided uncropped/unadjusted images of gels/blots in supporting information (Fig.3C-raw images, Fig.S1B-raw images and Fig.S1C-raw images) to meet the journal requirements. We also included a pdf file for labelled gels/blots named as “labelled gels/blots”.

The changes in the manuscript have been marked as track changes in different sections of the manuscript and named as “revised manuscript with track changes”. The specific responses to the reviewers’ comments are detailed in the “Response to Reviewers” section below. An unmarked manuscript file with accepted track changes was also provided and named as “Manuscript”. Graduate student (N.Ö.) performed revision experiments and her name was included as co-author of the study.

We hope that the updated manuscript will suffice to render this manuscript qualified for publication in your journal, and look forward to your decision.

Best regards,

Nihal Karakaş, M.Sc., PhD.

Department of Medical Biology,

İstanbul Medipol University

…

Regenerative and Restorative Medicine Research Center (REMER),

Research Institute for Health Sciences and Technologies,

İstanbul Medipol University 

Response to Reviewers:

Reviewer #1: 

Query: In this paper authors showed that bone marrow hMSCs either from cell line or healthy donors differentiate into functional neurons by treating them with induction medium.

The study is interesting but the paper needs some changes. In figures 1C, 3H and 4A immunofluorescence is not clear: authors should present fluorescence of the single proteins and DAPI in color (not only the merged panels).

Authors should better organize the paragraphs and the figure legends. In my opinion, they should put the figure legends at the end of the paragraph or of the paper.

Authors should evaluate the expression of proteins (βIII tub, Nestin, NSE. NF) at the same time points of PCR through WB analysis.

Response : We would like to thank the reviewer for the constructive comments.

We have now presented the fluorescence images in color and formatted figure legends have been presented according to the Plos One journal manuscript submission guide line below.

<Place figure captions in the manuscript text in read order, immediately following the paragraph where the figure is first cited. Do not include captions as part of the figure files or submit them in a separate document>.

We have also evaluated the expression of βIII tubulin and NSE by western blotting and these results are presented in Fig. S2-C. Protocol is placed in Methods section. As ordering and receiving antibodies from Turkey is a long process, we are still waiting to recieve other antibodies and therefore could provide data on βIII tubulin and NSE expression.

Additionaly, RT-PCR data in the manuscript was from hMSC donors. We performed western blot experiments on hMSC cell line used in the study since we could not receive any bone marrow sample during the revision period. Normally, we use bone marrow hMSCs freshly at passage#3 to 5 for neuronal differentiation studies. As the cell line expression profile is partially different than donor derived hMSCs, we also evaluated mRNA levels and included RT-PCR results of the same markers (βIII tubulin and NSE) in Fig. S2-B.

Reviewer #2: 

The study aims to clarify a difficult issue in the research on stem cells related to cell-based therapy/neurological disease modelling, namely the induction and maintenance of a neuron-like phenotype and neuronal functions. In the study, both cells from a bone marrow-derived human mesenchymal stem cell line and bone marrow-derived mesenchymal stem cells isolated from human donors, submitted to the established ‘single step cytokine-based induction protocol’, are demonstrated to be able to display a neuronal immunophenotype and functional activities of differentiating neurons. The Methodological and ‘Results’ sections are correct, the experimental protocols are described in detail and the results are well described and organized. Homwever, the two main points of the study i.e. the newly-designed protocol and the demonstration of neuron-like functions, are not adequately introduced and discussed.

Major Compulsory Revisions

Query: The Abstract, Introduction, and Discussion are incomplete or partly unclear. As an example:Introduction - The Introduction describes the aims of the work in an incomplete fashion. It does not adequately introduce the previous advances in the field. Moreover, if it is true that dozens of compounds have been described, and in various combinations, to achieve phenotypical traces of neurons, in this manuscript the style adopted for introducing these attempts is confusing. Please rewrite.

Response: Thank you for pointing this out. We have now revised the Abstract and Introduction based on reviewers suggestions.

The same criticism can be applied to the comments about the comprehension of the induction underlying mechanisms.At the end, although several data have been reported, the more recent and specific studies and methods in the field are not highlighted. Please completely rewrite.

Response: We have now revised the Introduction and also highlighted the specific studies and methods in field as suggested by the reviewer.

Discussion – A sufficient discussion is followed by a clear conclusion. Nevertheless, this reviewer has one substantial point of criticism and would like to encourage the Authors to consider this aspect in order to strengthen the entire study. Because the ability of MSCs to differentiate into cells of the neuronal lineage has been largely questioned, a more detailed, point-by-point comparison between the more recent researches, that demonstrated a successful and stable neuronal differentiation (also from a functional point of view), and the advantages suggested in the study, should be introduced.

Response: Based on reviewers‘ suggestion, we have now revised the Discussion section of the manuscript.

Minor Essential Revisions

Abstract - At the beginning of the abstract, the meaning of the sentence ‘Mesenchymal stem cells are one of the promising tissue specific stem cell source’, is not clear. Please clarify. In addition, the definition ‘neural protein expressing cells’ only appears in the abstract, and is not used in the text, even when appropriate, and the same can be said for ‘neuron-like cells’ that should be used in the correct context. With the same objective, standard acronyms need to be uniformly and appropriately used throughout the text.

Please carefully control for incorrect statements, inaccuracies, and typing errors throughout the manuscript.

Response: Based on reviewers‘ suggestion, the abstract, introduction and discussion parts have been extensively revised.

---

## [Decision Letter · Decision Letter 1]

7 Apr 2020

Neurons from human mesenchymal stem cells display both spontaneous and stimuli responsive activity

PONE-D-19-34566R1

Dear Dr. Karakas,

We are pleased to inform you that your manuscript has been judged scientifically suitable for publication and will be formally accepted for publication once it complies with all outstanding technical requirements.

With kind regards,

Gianpaolo Papaccio, M.D., Ph.D.

Academic Editor

PLOS ONE

Additional Editor Comments (optional):

Reviewers' comments:

Reviewer's Responses to Questions

**Comments to the Author**

1. If the authors have adequately addressed your comments raised in a previous round of review and you feel that this manuscript is now acceptable for publication, you may indicate that here to bypass the “Comments to the Author” section, enter your conflict of interest statement in the “Confidential to Editor” section, and submit your "Accept" recommendation.

Reviewer #1: All comments have been addressed

Reviewer #2: All comments have been addressed

2. Is the manuscript technically sound, and do the data support the conclusions?

Reviewer #1: (No Response)

Reviewer #2: (No Response)

3. Has the statistical analysis been performed appropriately and rigorously? 

Reviewer #1: (No Response)

Reviewer #2: (No Response)

4. Have the authors made all data underlying the findings in their manuscript fully available?

Reviewer #1: (No Response)

Reviewer #2: (No Response)

5. Is the manuscript presented in an intelligible fashion and written in standard English?

Reviewer #1: (No Response)

Reviewer #2: (No Response)

6. Review Comments to the Author

Reviewer #1: (No Response)

Reviewer #2: (No Response)

7. PLOS authors have the option to publish the peer review history of their article (what does this mean?). If published, this will include your full peer review and any attached files.

Reviewer #1: No

Reviewer #2: No

---

## [Editor Report · Acceptance letter]

21 Apr 2020

PONE-D-19-34566R1 

Neurons from human mesenchymal stem cells display both spontaneous and stimuli responsive activity 

Dear Dr. Karakas:

I am pleased to inform you that your manuscript has been deemed suitable for publication in PLOS ONE. Congratulations! Your manuscript is now with our production department. 

With kind regards,

on behalf of

Prof. Gianpaolo Papaccio 

Academic Editor

PLOS ONE